

# Mowing wet meadows reduces the health of their snail communities

Roland Farkas[1,2,3], Miklós Bán[3] and Zoltán Barta[3]

[1] Bükk National Park Directorate, Eger, Hungary
[2] Juhász-Nagy Pál Doctoral School, University of Debrecen, Debrecen, Hungary
[3] HUN-REN-DE Behavioural Ecology Research Group, Department of Evolutionary Zoology and Human Biology, University of Debrecen, Debrecen, Hungary

## ABSTRACT

Wet meadows harbor rich biodiversity, making them pivotal ecosystems worldwide. These habitats are commonly used for grazing or hay production for livestock. However, regular mowing can influence these habitats, potentially leading to significant repercussions for the animals residing within them. In order to investigate the effects of land management practices, we conducted an experimental study to compare snail communities in mowed and unmowed wet meadows in northern Hungary. We found that overall, mowing decreases snail populations, as well as species richness and diversity. Thus, our results suggest that routine mowing of wet meadows is deleterious to their snail communities. Based on these results, we suggest that designated patches of meadows that are regularly managed should be left uncultivated in their natural state. These patches can serve as potential colonization sites, facilitating the restoration of the entire meadow's ecological balance.

## INTRODUCTION

Wet meadows, which have at least some water cover for part of the year, are important habitats for their high biodiversity and transition role between permanently flooded habitats and dry grasslands. However, they are increasingly affected by both climatic changes and human activities. Climate change presents various future scenarios, the majority of which are unfavorable for maintaining natural habitats (*Joyce, Simpson & Casanova, 2016*). In turn, human activities vary widely, ranging from sustainable farming practices to the complete transformation of natural habitats. Together, these processes have significant implications, particularly for sensitive areas such as wetlands.

Wet meadows, akin to other wetland types, have experienced a substantial reduction in area across Europe over the past centuries (*Zedler & Kercher, 2005*). Presently, they are mostly used for agricultural purposes, and so their survival largely depends on appropriate management practices. Even undisturbed wet meadows require periodic intervention due to their susceptibility to invasive species (*Zedler & Kercher, 2004*).

Grazing and mowing are the primary methods of agricultural management. However, the outcomes of these regular management practices cannot be universally applied to all habitats (*Oelmann et al., 2009*). For instance, the response of indicator taxa or abiotic

Corresponding author
Roland Farkas, farkasro@yahoo.com

variables to the same management practice can differ across distinct habitats, making it inappropriate to extrapolate findings from dry meadows to wet meadows (*Oelmann et al., 2009*). Due to a greater emphasis on studying the effects of various land use practices in dry and semi-dry habitats, such as hay meadows (*e.g.*, *Bakker, Elzinga & Vries, 2002*; *Moog et al., 2002*; *Kormann et al., 2015*), compared to wet meadows (*Pech et al., 2015*), there is a need to conduct further research in water-dominated habitats.

In order to effectively monitor the impact of management practices, it is necessary to investigate diverse and sensitive species groups that serve as reliable indicators of changes and exhibit rapid responses to different management techniques (*Plantureux, Peeters & McCracken, 2005*). Invertebrates are particularly important members of communities, occupying various niches, thereby making them well-suited for assessing the effects of management practices (*e.g.*, butterflies: *Johst et al., 2006*; *Konvicka et al., 2008*; *Dover et al., 2010*; dung beetles: *Frank et al., 2017*; snails: *Książkiewicz, 2014*; *Wehner et al., 2019*; orthopterans: *Chisté et al., 2016*; spiders: *Szmatona-Túri et al., 2017*).

The litter layer, which exists just above the soil surface, is an important component of wetland habitats, as processes occurring here are critical for the nutrient cycle. For instance, it is here that detritivores decompose the majority of deceased plant matter. Gastropods, commonly known as snails, play a significant role in the decomposition processes (*Newell, 1967*). Snails facilitate the activity of microbial detritivores by shredding materials, and their excretion of feces and production of mucus create favorable conditions for the proliferation of microbial life (*Theenhaus & Scheu, 1996*). Wetland habitats, including the litter layer and the vegetation layers immediately above it, often harbor diverse snail communities. Due to their limited dispersal capabilities and the high water content in their bodies, snails are highly susceptible to changes in environmental conditions. Consequently, they serve as excellent indicators of the prevailing conditions within these habitats (*Čejka & Hamerlík, 2010*; *Pech et al., 2015*; *Wehner et al., 2019*). Studies have indeed demonstrated that snail assemblages in wet meadows are strongly influenced by abiotic factors such as moisture, pH, and calcium content (*Martin & Sommer, 2004a*; *Cernohorsky, Horsák & Cameron, 2010*; *Hettenbergerová et al., 2013*; *Horsák, Zelený & Hájek, 2014*; *Wehner et al., 2019*). Different land-use practices can directly impact snail assemblages, with changes in abiotic factors resulting from management activities often leading to alterations in species richness and abundance (*Wehner et al., 2019*).

Here, we investigated how land management, specifically mowing, affects the snail communities of wet meadows. The hilly counties in northern Hungary have a diverse range of land-use practices within a mosaic structure, due to the variety of topographical and environmental factors. Our research focused on sites in the Putnok Hills and Cserehát Hills, characterized by narrow valleys traversed by small streams. To assess the effects of mowing, we established designated treatment and control plots within the managed and unmanaged regions of each valley, respectively. Consequently, we evaluated the effects of the management practices by comparing the snail communities inhabiting pairs of managed and control plots.

## MATERIALS AND METHODS

### Study area

The study was conducted within the Putnok Hills and Cserehát Hills, in northern Hungary, Europe (Fig. 1). The elevations of the hills surrounding the valleys in our study range from 250 to 350 m above sea level (asl), while the valleys themselves are situated at approximately 220 to 250 m asl. The study area falls within the temperate continental climatic zone, characterized by an average annual temperature of 8.5 °C to 9 °C and an average annual precipitation ranging between 550 to 600 mm (*Mezősi, 1998*; *Dobány, 2010*).

Experimental plots, comprising both managed and control conditions, were established within the study sites. The size of the plots was in all cases larger than 1 hectare. The managed plots were within areas subjected to mowing. In close proximity to each managed plot, at an average distance of 204.8 ± 117.7 (SD) meters, a corresponding control plot was established in unmanaged areas. All plots were positioned at the valley bottoms, close to small streams. Each pair of control and managed plots shared the same valley floor, ensuring comparability between the two conditions.

All experimental plots were wet meadows and they were completely devoid of shrubs or willows throughout the duration of the study. Typically, the valley bottoms had a predominance of sedge marsh vegetation. Since the area had not been actively cultivated for several decades, the sedge species *Carex acutiformis* and reed (*Phragmites australis*) emerged as the predominant plant species, covering the study areas in similar proportions. Grasses and forbs were present in all plots; however, their distribution was moderate, and they did not occupy a substantial portion of the surface. Agricultural management was started in 2003, when both managed and control plots were established at the bottom of each valley in our study area. Initially, there were no discernible differences between the vegetation of the managed and control plots. The managed plots were mowed annually with a tractor, with the cut plant material removed. The control plots were completely unmowed since 2003, and there was no management of the vegetation growing there. Consistent methodologies were employed for all managed sites, ensuring uniformity in the management practices.

### Sampling design

To sample the experimental plots, within each one we placed five sampling quadrats measuring 25 by 25 cm. These quadrats were equidistantly positioned along a 50-m transect (Fig. 1). At our seven study sites, a total of 15 transects were established, with eight located within managed plots and seven within control plots. At the Buda-völgy site (no. 2), two managed plots were set up due to the presence of two mowed plots on the same valley floor as the control plot. The sampling was conducted in 2007 and 2008, in July and August, after the completion of the management activities. The transects were positioned in the same locations in both years. Our total number of sampled quadrats in the study was 150.
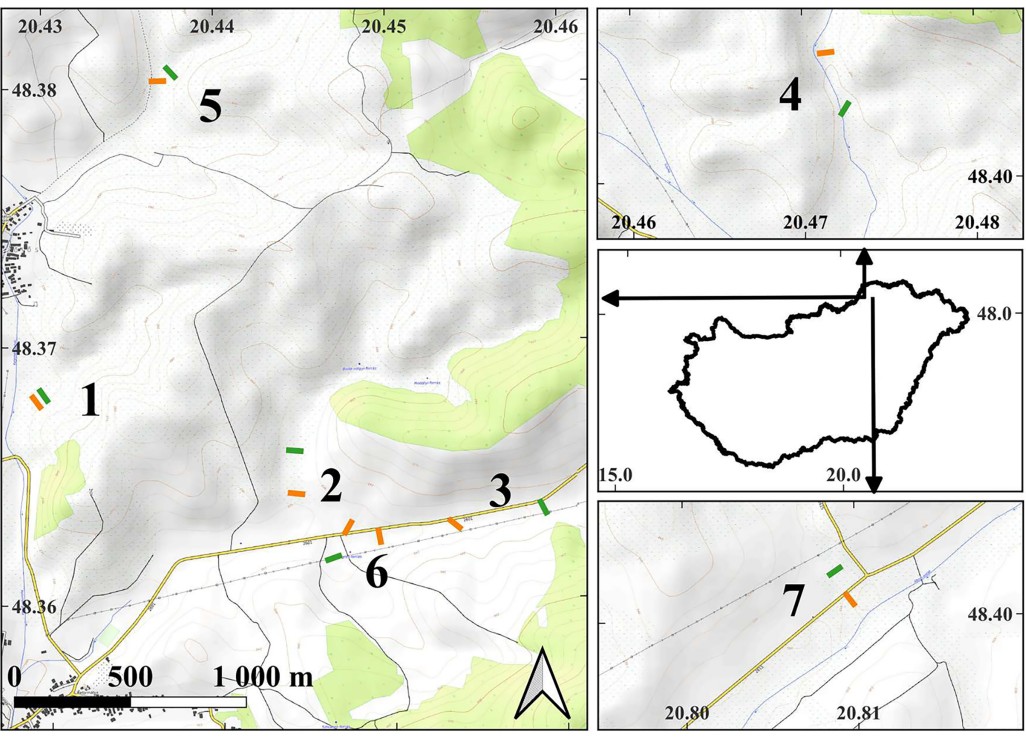

**Figure 1 The map of the study areas.** On the middle of right side, the contour of Hungary is shown with arrows indicating the locations of the detailed maps within the country. Short lines indicate the locations of the transects in the examined plots: orange lines—managed plots, green lines—control plots. The depicted length of the transects is proportional to their actual length. The numbers next to the transects indicate the serial numbers of the study sites. The detailed maps were created using the OpenTopoMap (opentopomap.org) base map. Map data: © OpenStreetMap contributors, SRTM | Map display: © OpenTopoMap (CC-BY-SA). All detailed maps share the same scale and orientation. WGS84 geographic coordinates are marked on the axes.

Sampling procedures involved the complete removal of litter and the top 1 cm layer of soil from each quadrat. These samples were collected and stored in plastic bags. The snail shells (*Gastropoda*) were manually separated from the litter samples using a delicate nipper and subsequently identified under a stereomicroscope. Identification protocols followed the guidelines of *Kerney, Cameron & Jungbluth (1983)* and *Glöer & Meier-Brook (2003)*. Only fresh shells, characterized by intact whorls and non-eroded periostracum, were selected for identification. All fresh shells were identified at the species level. Slugs were not included in the survey due to the unsuitability of this method for accurate quantitative estimation of their abundance, as noted by *Cameron & Pokryszko (2005)*.

## Statistical analyses

To analyze the impact of management practices, we first employed a multilevel modeling framework similar to *Jackson et al. (2012)*. This framework enabled us to examine the effects of environmental factors on both the entire community and individual species. It involves both fixed and random terms, for both intercept and slope. The fixed terms describe the behavior of the entire community (that is, the average of all species involved) and the random terms (which specify a distribution centered around the fixed terms)

describe the variability of that behavior between species. Thus, a model with a negative fixed effect and a small standard deviation implies a negative difference between treatment conditions, with a similar effect for all species. In contrast, a negative fixed effect and a large standard deviation imply a negative difference on average but large variation between species.

We constructed three such models, to examine species populations, the presence of each species and Shannon diversity in the experimental plots. In the first, the response variable was the number of individuals of each species observed in a sample quadrat. Fixed effects included the type of management, year, and their interaction. Random effects included a random intercept for plot ID nested within site ID, a random intercept within species ID, and random slopes for management and years within each species. Considering the apparent over-dispersion in the data, a negative binomial error distribution was employed. Model fitting was performed in all cases using the 'glmmTMB' function from the 'glmmTMB' package (*Brooks et al., 2017*).

In our second model, the response variable was the presence/absence of each species in a sample quadrat. The model structure was the same as described above, except here we used a binomial error distribution.

Finally, we investigated the impact of management on biodiversity by calculating the Shannon-Weaver diversity index (*Shannon & Weaver, 1949*) for each sample quadrat. This was the response variable in our third model. Fixed effects included the type of management, year, and their interaction. Random effects were a random intercept for plot ID nested within site ID. We did not include species-related random terms since species identity was not relevant in this case.

Following the fitting of the full models, we conducted tests to examine the effects of random terms by successively removing each random term and comparing the models using likelihood-ratio tests performed with the 'anova' function in R (Table 1). Because it was not significant, we removed the random slope for year from the full model (Table 1). We assessed the significance of fixed terms by using the 'summary' function in R. We checked how the fitted models fulfill their assumptions by analyzing their residuals with the DHARMa R package (*Hartig, 2022*). All of our model fitted to the data well.

As estimates of species richness and diversity measures can strongly depend on sampling efforts (*Gotelli & Colwell, 2001*) we also used an approach based on rarefaction analysis (*Chao et al., 2014*; *Hsieh, Ma & Chao, 2016*). To calculate different measures of diversity we used the 'iNEXT' R package (ver. 3.0.0, *Hsieh, Ma & Chao, 2022*). As year turned out to not be significant we grouped data from both years and only compared the effect of management on species diversities. Note, the iNEXT package cannot take into account the random effect structure of our experiment (*i.e.*, plots paired within areas), therefore we present the results of both the multilevel models and rarefaction analyses.

All species collected were included in the statistical analyses. All statistical analyses were carried out in the R interactive statistical environment (version 4.2.2, *R Core Team, 2022*).

## RESULTS

We collected 11,629 specimens belonging to 34 snail species (Fig. 2).

**Table 1 Model selection of multilevel model for species abundance.**

| Summary statistics | Random terms | Models | | | | |
|---|---|---|---|---|---|---|
| | | Full | -Year slope | -Treatment slope | -Species intercept | -Plot/site intercept |
| Standard deviation | Plot/site intercept | 0.23 | 0.23 | 0.23 | 0.19 | – |
| | Site intercept | 0.23 | 0.23 | 0.22 | 0.18 | – |
| | Species intercept | 1.84 | 1.84 | 1.87 | – | 1.80 |
| | Year slope | 0.03 | – | 0.02 | 1.57 | 0.00 |
| | Treatment slope | 0.37 | 0.37 | – | 1.62 | 0.35 |
| AIC | | 11,233 | 11,231 | 11,256 | 11,989 | 11,342 |
| $\chi^2$ | | | 0.00 | 24.65 | 757.79 | 112.98 |
| $p$-value | | | 0.95 | **0.000** | **0.000** | **0.000** |

Note:
The response variable is the number of individuals of each species within a sample. The full model contains fixed terms for treatment, year and treatment x year, random intercepts for plot within each site and species, and random slopes for treatment and year within each species. The table shows the effects of removing each random term (intercept or slope) from the full model. It lists the standard deviations attributed to each remaining random term, as well as the summary statistics of that model. The AIC values are the Akaike Information Criteria values for each model, the $\chi^2$ values are the test statistics obtained by comparing the given model to the full model with likelihood ratio tests, while the $p$-values indicate the level of significance of the test statistics. The degree of freedom is one for all $\chi^2$ except for the '-plot/site' model where it is two. Significant results are highlighted in bold.

In our multilevel model for species abundance, we found that all random terms had significant effects, including the random slope for management (Table 1). Moreover, the only significant fixed term was that for management (Table 2). Put together, this indicated (1) that different species reacted differently to mowing (Fig. 3). In fact, the populations of 21 species responded more strongly to mowing (negative random slope) than the average, while the opposite was true for 13 species (Fig. 3). However, (2) despite these variable responses, the entire snail community declined due to mowing (Table 2).

Similar to species abundance, the presence/absence of species exhibited substantial variability both among areas and species, and there was significant variation in species-specific responses to management practices (Table 3). The only significant fixed effect was management, indicating that again, while there was substantial variation between species, on average there were fewer species in managed areas (Table 2).

Our model for Shannon diversity revealed that the variation among plots and areas was not statistically significant ($\chi^2_2 = 2.00$, $p = 0.367$). Among the fixed terms, the effect of management was significant, again indicating that snail communities exhibited lower diversity in managed areas (Table 2).

The rarefaction analyses support the results of the multilevel models, as in all three diversity measures the mowed areas have lower values compared to the controls (Fig. 4). This indicates lower diversity in the mowed areas, whatever the sample size of our experiment. These differences were significant in cases of both the Shannon and the Simpson diversity indices (the confidence intervals do not overlap). In the case of species richness, the confidence intervals overlapped considerably, so the difference between the control and mowed areas was not significant.

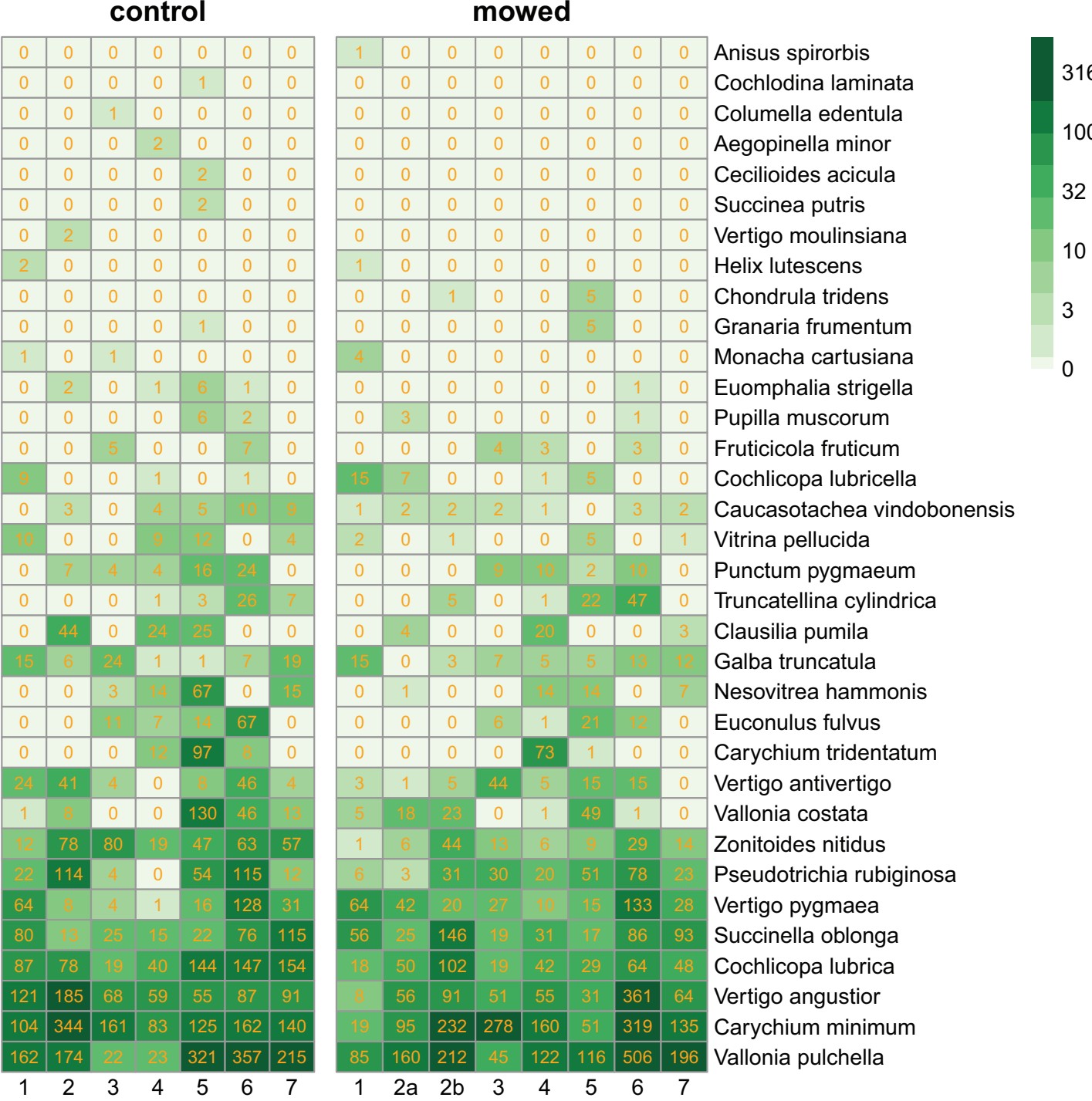

**Figure 2 Heatmaps of control and mowed plots.** Species (in rows) follow each other in ascending order according to their abundance. Columns correspond to plots within sites as indicated in Fig. 1. Cells contain the number of individuals in each plot. The green scale emphasizes the species abundance.

**Table 2 Multilevel model results.**

| Model component | | Species abundance | | | Species presence | | | Shannon diversity | | |
|---|---|---|---|---|---|---|---|---|---|---|
| | | Parameter estimate | Standard error | Confidence interval | Parameter estimate | Standard error | Confidence interval | Parameter estimate | Standard error | Confidence interval |
| Fixed terms | (Intercept) | −0.50 | 0.35 | −1.19, 0.19 | **−1.62***** | **0.41** | **−2.42, −0.82** | **0.80***** | **0.03** | **0.75, 0.85** |
| | Year | 0.12[+] | 0.07 | −0.01, 0.26 | −0.09 | 0.12 | −0.32, 0.15 | −0.04 | 0.03 | −0.1, 0.03 |
| | Management | **−0.40***** | **0.17** | **−0.73, 0.07** | **−0.55***** | **0.22** | **−0.98, −0.11** | **−0.01****** | **0.03** | **−0.16, −0.03** |
| | Year × Management | 0.07 | 0.1 | −0.12, 0.27 | 0.226 | 0.17 | −0.11, 0.56 | 0.04 | 0.04 | −0.05, 0.12 |
| Random terms | SD (Intercept plot/site) | 0.23 | | | 0.29 | | | 0.03 | | |
| | SD (Intercept site) | 0.23 | | | 0.18 | | | 0.01 | | |
| | SD (Intercept species) | 1.84 | | | 2.15 | | | | | |
| | SD (management slope species) | 0.37 | | | 0.52 | | | 0.13 | | |
| Dispersion | (Intercept) | 11.04 | | 9.95, 12.26 | | | | | | |
| | Num.Obs. | 5,100 | | | 5,100 | | | 150 | | |
| | AIC | 11,231.5 | | | 654,395 | | | −156.3 | | |

**Notes:**

The table shows the parameter estimates, their standard errors and confidence intervals of our models for species abundance, presence and Shannon diversity (for the full model without a random slope for year).

*** Marks $p < 0.001$.

** Marks $p < 0.01$.

* Indicates $p < 0.05$.

+ Indicates $p < 0.1$.

The negative parameter estimate for treatment implies a lower abundance in mowed areas. The standard deviations for each random term are also given, as well as the model AICs. Significant results are highlighted in bold.

## DISCUSSION

We found that regular mowing negatively impacted on all the assessed characteristics of the snail communities, including population sizes and community diversity. These findings align with previous studies that have reported similar outcomes. For instance, *Pech et al. (2015)* observed decreased snail abundance and species richness on mowed plots within wet meadows. *Chisté et al. (2016)* also documented comparable results for orthopterans. The intensive mowing of wet meadows also diminishes slug abundance (*Everwand, Scherber & Tscharntke, 2013*). Moreover, similar negative effects on abundance have been observed in other taxa as well, such as spiders (*Cattin et al., 2003*) and dung beetles (*Frank et al., 2017*).

Two distinct analysis methods were used to examine the change in diversity: the multilevel modeling framework used by *Jackson et al. (2012)* and rarefaction analysis (*Chao et al., 2014*; *Hsieh, Ma & Chao, 2016*). Both methods led to similar results. The species richness decreased in both cases as a result of the treatment, but the difference was only significant when the multilevel modeling framework was applied. In the case of Shannon diversity, both approaches showed a significant decrease. This was also confirmed by examining Simpson diversity using rarefaction analysis.

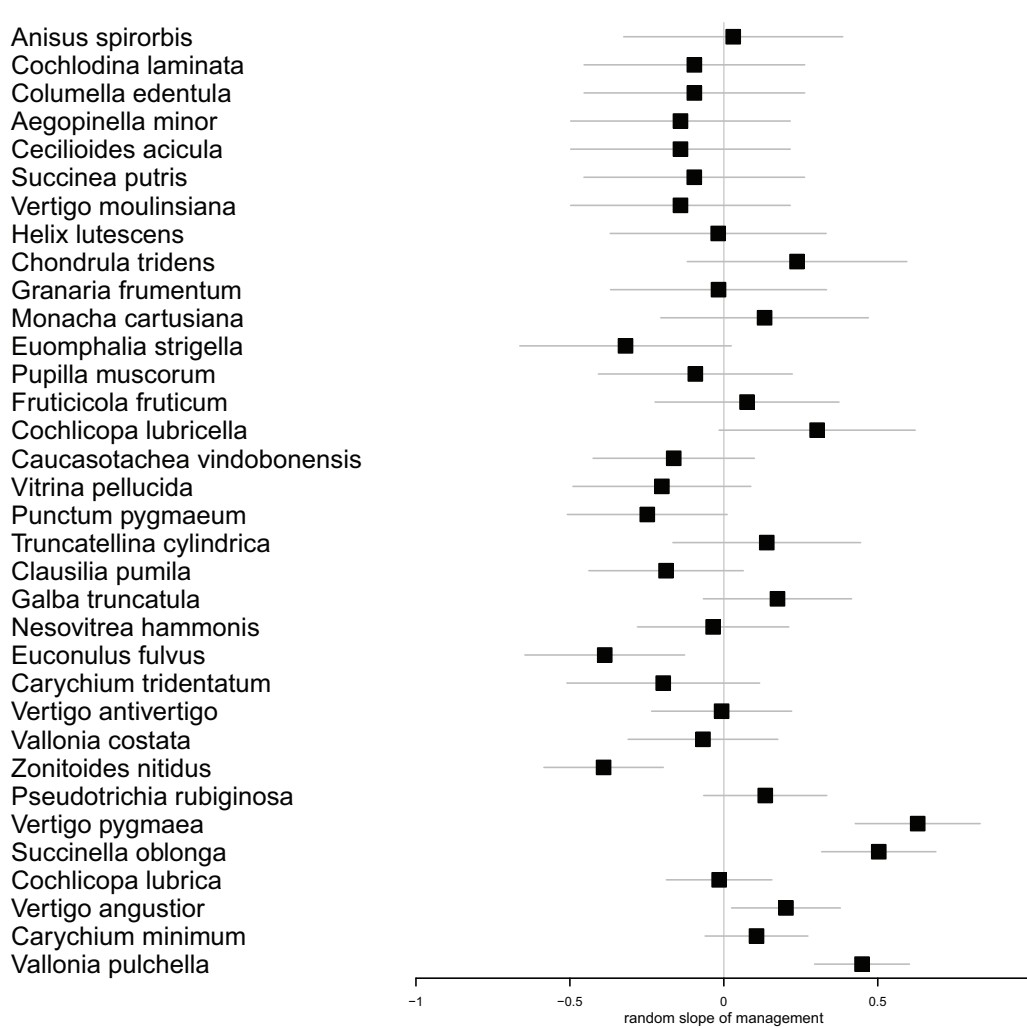

**Figure 3 Forest plot showing the individual estimates of random slopes for management for each species.** Horizontal error bars indicate standard errors of the estimates. Species with positive slopes responded less to mowing. The species follow each other in ascending order according to their abundance.

Our study indicates that some of the adverse effects associated with regular mowing can be detected within a relatively short period of its start (4–5 years in our case). It should be noted that longer-term treatments in the examined area may impact additional community attributes.

Abiotic characteristics of the habitat, such as moisture levels and litter thickness, are believed to influence snail communities (*Martin & Sommer, 2004b*; *Dvořáková & Horsák, 2012*; *Pech et al., 2015*). Therefore, it is plausible that management practices modify snail communities by altering these habitat traits. For instance, studies have demonstrated that vegetation mowing leads to increased direct radiation, temperature, and decreased soil moisture (*Lepš, 1999*; *Zechmeister et al., 2003*). However, in certain cases, when the abiotic factors remain in good condition, they may mitigate the worst effects of mowing and still

**Table 3 The effect of random terms on species presence.**

| Summary statistics | Random terms | Models | | | | |
| --- | --- | --- | --- | --- | --- | --- |
| | | Full | -Year slope | -Treatment slope | -Species intercept | -Plot/area intercept |
| Standard deviation | Plot/site intercept | 0.29 | 0.29 | 0.29 | 0.22 | |
| | Site intercept | 0.18 | 0.18 | 0.19 | 0.13 | |
| | Species intercept | 2.15 | 2.15 | 2.19 | | 2.11 |
| | Year slope | 0.00 | | 0.00 | 1.78 | 0.00 |
| | Treatment slope | 0.52 | 0.52 | | 1.98 | 0.51 |
| AIC | | 3,693 | 3,691 | 3,704 | 4,311 | 3,727 |
| $\chi^2$ | | | 0.00 | 12.23 | 619.94 | 37.93 |
| p-value | | | 1.000 | **0.000** | **0.000** | **0.000** |

Note:
The response variable is the number of species within a sample. The full model contains fixed terms for treatment, year and treatment x year, random intercepts for plot within each site and species, and random slopes for treatment and year within each species. The table shows the effects of removing each random term (intercept or slope) from the full model. It lists the standard deviations attributed to each remaining random term, as well as the summary statistics of that model. The AIC values are the Akaike Information Criteria values for each model, the $\chi^2$ values are the test statistics obtained by comparing the given model to the full model with likelihood ratio tests, while the p-values indicate the level of significance of the test statistics. The degree of freedom is one for all $\chi^2$ except for the '-plot/site' model where it is two. Significant results are highlighted in bold.

provide tolerable conditions for snails. The effect of management was decisive and exerted a stronger influence than the potential impact of abiotic factors.

Regular management practices have the potential to reduce the abundance of vulnerable and specialist species, even when their numbers are already below critical levels (*Książkiewicz, 2014*; *Kormann et al., 2015*). In our case, we did not directly observe this phenomenon. However, we found that different species exhibited varying responses to the treatment. Among the species that showed the most significant decrease in population size (*Zonitoides nitidus, Vitrina pellucida, Clausilia pumila, Euomphalia strigella*), we observed species that prefer both wetter and drier habitats (*Welter-Schultes, 2012*). On the other hand, the species that responded to mowing less intensively (*Cochlicopa lubricella, Truncatellina cylindrica, Vertigo pygmaea, Chondrula tridens*) typically favor drier and more open habitats, demonstrating a greater tolerance to disturbance. While meadow management offers economic benefits, it is crucial to prioritize the preservation of ecological values (*Joyce, Simpson & Casanova, 2016*; *Felipe-Lucia et al., 2020*). This is particularly important in the context of wet meadows, which have high ecological value, as well as in habitats housing vulnerable taxa.

Given the diverse requirements of different invertebrate taxa, finding suitable management methods is a challenging task. Therefore, it is worthwhile to consider a patchy composition of management at both the landscape and managed unit levels (*Pech et al., 2015*). In the context of our study, this entails leaving unmowed patches within wet meadows to ensure the survival of diverse snail communities, which can then recolonize the mowed areas after each annual mowing. These areas are expected to undergo ecological succession over the long term. Our study focused on abandoned areas where shrub growth had not yet commenced and was not observed until the end of the project. Consequently, the response of the studied snail community to slow succession could not be assessed within the relatively short duration of the study.

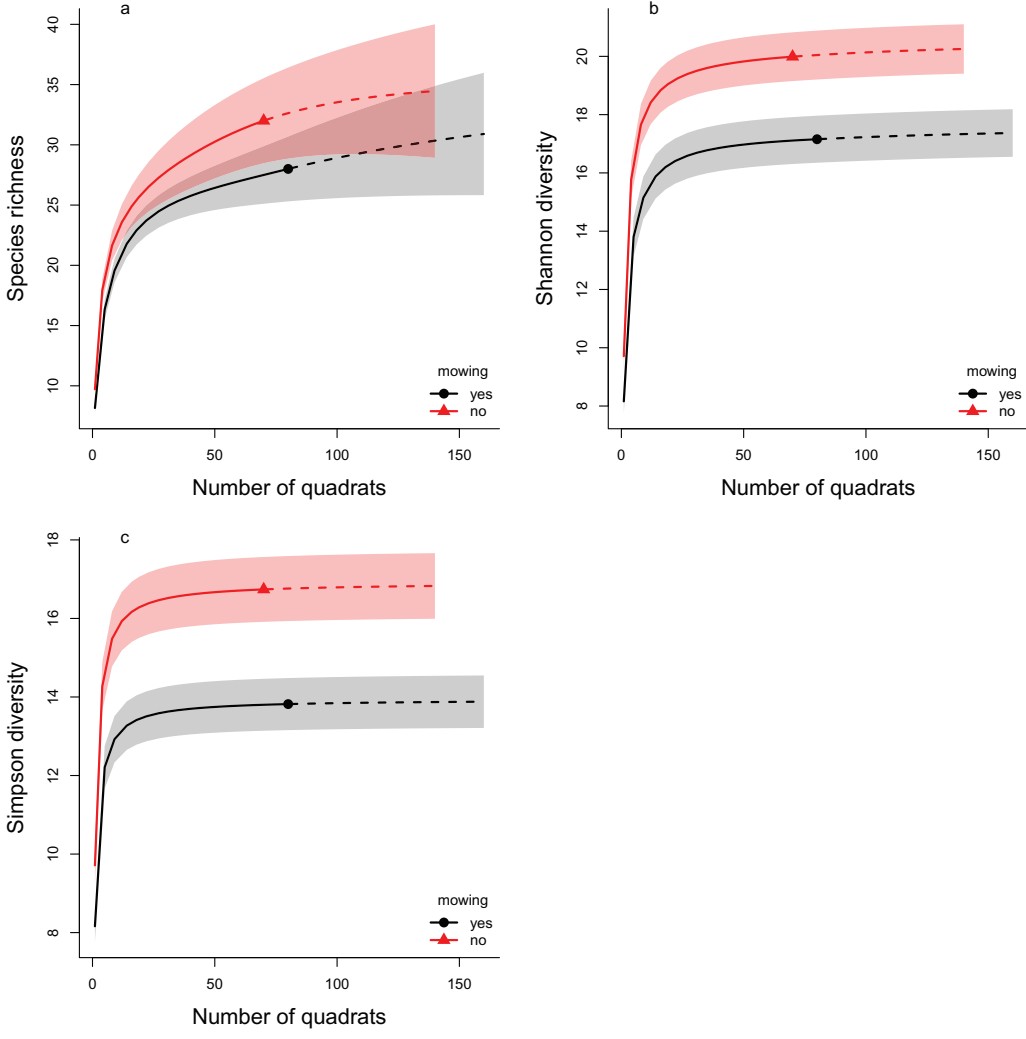

**Figure 4 The rarefaction analyses of the effect of mowing on Hill number based diversity measures.**
The plotted diversity measures are (A) species richness, (B) Shannon diversity and (C) Simpson diversity. The continuous lines show the interpolated, while the dashed lines the extrapolated estimates. The symbols (filled triangle: control, filled circle: mowed) mark the observed values. The shaded areas show the confidence intervals around the curves. Red color indicates the control areas, while black the mowed ones. Non-overlapping regions indicate significant difference.

To address the challenges posed by management practices, alternative approaches can be explored. The specific implementation details of management, such as timing, equipment used, and the handling of mowed biomass, can have varying effects on plant and invertebrate communities (*e.g.*, *Humbert et al., 2010*, *2012*), particularly when considering the specific traits of the habitats involved. Conducting further studies on the interaction between management practices and local habitat traits is crucial for gaining a comprehensive understanding of the underlying mechanisms operating in managed wet meadows. This research will contribute to identifying appropriate strategies for the sustainable management and cultivation of wet meadow ecosystems.

## CONCLUSIONS

Wet meadows represent fragile ecosystems, and alterations in snail communities serve as reliable indicators of their ecological status. In this study, we conducted experimental investigations to assess the impact of mowing on snail communities inhabiting wet meadows. Through a balanced experimental design, we confirmed that mowing exerted negative effects, leading to reductions in species abundance, species presence, and Shannon diversity at the community level. Mowing conducted in wet habitats appeared to be favorable for snail species typically associated with drier habitats. This suggests that mowing constitutes a transformation of the original habitat. Over the long term, the negative effects of mowing could potentially lead to even less favorable conditions for the original ecosystems. These findings emphasize the importance of considering these detrimental effects when formulating management strategies for wet meadows. Therefore, we recommend maintaining unmowed patches in wet meadows that are regularly managed, in order to maintain their original ecosystems.

## ACKNOWLEDGEMENTS

The authors would like to express their sincere gratitude to Barna Páll-Gergely for providing valuable feedback and to Bernadett Virókné Fodor for their assistance with linguistic corrections on a previous version of the manuscript. Special thanks are extended to György Dudás for his insightful suggestions regarding the interpretation of the results. We are grateful to Tamás Székely, Jr. for his help to correct our English.

### Funding

This was prepared with the professional support of the Doctoral Student Scholarship Program of the Co-operative Doctoral Program of the Ministry of Innovation and Technology financed from the National Research, Development and Innovation Fund (Hungary). The study was supported by a National Research, Development and Innovation Office grant (NKFIH K138503). Zoltán Barta was supported by project no. TKP2021-NKTA-32 which has been implemented with the support provided by the Ministry of Culture and Innovation of Hungary from the National Research, Development and Innovation Fund, financed under the TKP2021-NKTA funding scheme. The funders had no role in study design, data collection and analysis, decision to publish, or preparation of the manuscript.

### Grant Disclosures

The following grant information was disclosed by the authors:
Doctoral Student Scholarship Program of the Co-operative Doctoral Program of the Ministry of Innovation and Technology financed from the National Research.
Development and Innovation Fund (Hungary).
National Research, Development and Innovation Office Grant: NKFIH K138503.
Zoltán Barta: TKP2021-NKTA-32.

Ministry of Culture and Innovation of Hungary from the National Research.
Development and Innovation Fund.
TKP2021-NKTA.

## Competing Interests

The authors declare that they have no competing interests.

## Author Contributions

- Roland Farkas conceived and designed the experiments, performed the experiments, analyzed the data, prepared figures and/or tables, authored or reviewed drafts of the article, and approved the final draft.
- Miklós Bán analyzed the data, authored or reviewed drafts of the article, and approved the final draft.
- Zoltán Barta performed the experiments, analyzed the data, prepared figures and/or tables, authored or reviewed drafts of the article, and approved the final draft.

## Data Availability

The datasets generated and/or analysed during the current study are available at the University of Debrecen: Roland, Farkas; Bán, Miklós; Zoltán, Barta, 2022, "Effect of mowing on snail communities in wet meadows", https://doi.org/10.48428/ADATTAR/EYCRYY, University of Debrecen, V3, UNF:6:JvLNdnz4V70+9MsjQbev/A== [fileUNF].

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
