# Peer review of "Mowing wet meadows reduces the health of their snail communities"

_PeerJ, doi:10.7717/peerj.16783_

## Round 0.1 · original submission · Major Revisions

First of all, I'm sorry for the long delay for you to have a first decision on this. I have been recently assigned this manuscript.

We have two reviews of your manuscript. Both of them saw your paper with great enthusiasm and suggested only a couple of minor revisions. However, after taking a look at the manuscript myself, I also have many more comments and that's why I'm recommending major revisions:

1) Fig. 1 looks like a map, but it's actually not a map. Notice that it lacks key information, such as coordinates along axes of each inset. Figures have to stand by themselves. As such, I don't know what's the figure on the upright corner. It can be whatever a region, a country because there's no information on the figure legend to guide the reader. A scale is also lacking. An arrow point north is lacking. Are those contour lines really necessary? Do they dialog with the text, research question? What are those grey lines, roads?

2) A cartoon showing the distribution of control and managed sites is welcomed, and the plots within them. What is the size of each plot?

3) Delete Table 1, include lat-long coordinates along axes of Fig. 1

4) The legend of Fig. 2 is not correct. Those are not "traits" of communities, but properties. But see further comments on data analysis, You will have to replace these analyses all together.

5) All analysis on species diversity that compare control and managed sites have to be re-done. You can't compare the richness of two classes of sites without taking into account differences in sampling effort (Gotelli & Colwell 2001 Ecol Lett). As such, I highly recommend you to replace the comparisons of species richness, abundance and Shannon diversity index by the method that uses Hill numbers as implemented in the R package iNExt (see also Shiny app in Anne Chao's website). That way you can use the 95%CI of each q order to test for differences between the two types of sites in terms of effective number of species, rare and common species.

6) the MLM is akin of what we'd call today joint-species distribution models. It's fine to use glmmTMB, but the interpretation of the model is not correct. Also, *you need to diagnose the model residuals*. Consider using the R package DHARMa. Also, I'm not sure if including year is necessary. But more importantly, you need to provide a table and ideally a plot showing the results of the model (maybe a forest plot). See R packages performance and modelsummary. Fig 2 doesn't show the full output of the model, with random effects etc.

7) You only have year and treatment as fixed factors, so why use a stepwise procedure here? Also, both factors have only two levels, you don't need post hoc tests in this case

8) by definition you can't calculate the portion of variance accounted for random effects. The nearest you can get is calculating Nakagawa's R2 for marginal and conditional parts of the model

I have made some suggestions on the English language, but the text should undergo a careful revision for English grammar and writing style.

**Language Note:** The Academic Editor has identified that the English language must be improved. PeerJ can provide language editing services - please contact us at [email protected] for pricing (be sure to provide your manuscript number and title). Alternatively, you should make your own arrangements to improve the language quality and provide details in your response letter. – PeerJ Staff

Reviewer 1 ·

Basic reporting

The manuscript presents the results of the two-iyear study on the composition of snails' diversity and abundance as an effect of mowing.
English used in the paper is of good quality.
The literature references provided are of good value and well-suited to the content of the work.
The article has a proper structure, and the figures and tables are of quality.
The results presented in this study are relevant, and they confirm a hypothesis that regular moving affects communities of snails in the wet meadows. That is why the control, unmanaged plots are essential to maintain the communities of snails in natural areas.

Experimental design

The article is well suited to the aims of this journal.\
The research questions are well-defined. They are relevant and meaningful.
The investigation performed in this study is of good technical and ethical standards.
Methods are described with sufficient detail and information to replicate.

Validity of the findings

The findings are of significance to the field of study. All needed data have been provided.

However, the results are too concise. They should be described in more detail.

I was missing in this work information about snail species that were more susceptible to mowing and which - more resistant. This information could be extracted from Table S1.

Discussion lines 280-283 are not necessary for this work. Consider removing it or rephrasing it to fit the aim of your work.

The conclusions must be improved. They are too general. It would be more valuable if some details were added to that part of the work.

·

Basic reporting

Generally well written but it could use some copy editing. For example, in the abstract you talk about the "act of mowing" whereas you should simply say mowing.

Experimental design

no comment; this is well done

Validity of the findings

no comment; these are valid with a good statistical analysis

Additional comments

A good paper on a very understudied group of animals.

---

## Round 0.2 · accepted · Accept

Thank you for carefully revising the manuscript addressing all comments by my and reviewers.

During the proof checking phase, remember to replace ‘species population’ by ‘species abundance’ in L 153.

I’m glad to recommend it for publication as is. Congratulations

Reviewer 1 ·

Basic reporting

The revised version of the manuscript entitled "Mowing wet meadows reduces the health of their snail communities" is ready to be published.
The authors use clear and professional English language.
The article includes a sufficient introduction and background. Relevant literature is appropriately cited.
The article structure is correct. The graphs are clear and of good quality. They contain relevant content for this article.
Results are relevant to the field.

Experimental design

The article fits within the scope and aims of the PeerJ journal.
Research questions are well-defined.
The research performed was correct and of a very good standard.
Methods are described with sufficient details and information to replicate.

Validity of the findings

Conclusions are well stated and linked to results.

·

Basic reporting

looks good

Experimental design

fine

Validity of the findings

looks good

Additional comments

I have no further comments; it looks good to me. A nice contribution